# Basal ganglia-cortical connectivity underlies self-regulation of brain oscillations in humans

Kazumi Kasahara [1,2,3,5], Charles S. DaSalla [1,2,5], Manabu Honda[2] & Takashi Hanakawa[1,2,4,5✉]

Brain-computer interfaces provide an artificial link by which the brain can directly interact with the environment. To achieve fine brain-computer interface control, participants must modulate the patterns of the cortical oscillations generated from the motor and somatosensory cortices. However, it remains unclear how humans regulate cortical oscillations, the controllability of which substantially varies across individuals. Here, we performed simultaneous electroencephalography (to assess brain-computer interface control) and functional magnetic resonance imaging (to measure brain activity) in healthy participants. Self-regulation of cortical oscillations induced activity in the basal ganglia-cortical network and the neurofeedback control network. Successful self-regulation correlated with striatal activity in the basal ganglia-cortical network, through which patterns of cortical oscillations were likely modulated. Moreover, basal ganglia-cortical network and neurofeedback control network connectivity correlated with strong and weak self-regulation, respectively. The findings indicate that the basal ganglia-cortical network is important for self-regulation, the understanding of which should help advance brain-computer interface technology.

[1] Department of Advanced Neuroimaging, Integrative Brain Imaging Center, National Center of Neurology and Psychiatry, Tokyo 187-8551, Japan. [2] Department of Functional Brain Research, National Center of Neurology and Psychiatry, Tokyo 187-8551, Japan. [3] Human Informatics and Interaction Research Institute, National Institute of Advanced Industrial Science and Technology, Ibaraki 305-8566, Japan. [4] Integrated Neuroanatomy and Neuroimaging, Kyoto University Graduate School of Medicine, Kyoto 606-8501, Japan. [5] These authors contributed equally: Kazumi Kasahara, Charles S. DaSalla, Takashi Hanakawa. ✉email: hanakawa.takashi.2s@kyoto-u.ac.jp

Brain-computer interfaces (BCIs) provide an artificial link by which the brain can interact with the environment without using bodily effectors or sensors[1–4]. In BCIs, signals from the brain are retrieved and decoded to control external devices. Neurofeedback (NFB) is a related technique which encourages users to control their own brain activity according to decoded brain signals[5–8]. Potentials for real-world applications are emerging for both BCIs[9–13] and NFB[14–18], though these techniques have limitations. Brain networks of BCI/NFB users are engaged in fine-tuning their own neural states, involving self-regulation of brain activity or connectivity[7,19–22]. Furthermore, BCI/NFB performance varies across individuals, reflecting interindividual differences in the self-control of brain states[23–27]. Some participants fail to self-regulate brain activity, even after repeated training sessions[18,28,29].

Further development of BCI technology may overcome those limitations[30], but the issue will remain for NFB, which inherently relies on the ability to self-regulate brain activity (here, referred to simply as self-regulation). How organisms achieve self-regulation remains unknown, despite continued efforts to identify the underlying neural mechanisms and their correlates[18].

Self-regulation may involve the neurofeedback control network (NfCN), which includes the anterior insula cortex (AIC), anterior cingulate cortex, supplementary motor area (SMA), dorsolateral prefrontal cortex (dlPFC), lateral occipital cortex, and superior and inferior parietal lobules (SPL and IPL, respectively)[14,16]. The NfCN corresponds primarily to the cognitive control network[22], which has been implicated in top-down cognitive control. Thus, The NfCN might subserve the top-down control of self-regulation on the basis of explicit knowledge about the strategy (i.e., think strategy)[31]. Alternatively, the basal ganglia-cortical network (BgCN), which underlies behaviors stemming from trial-and-error-type procedural learning, may be the core neural correlate of BCI/NFB control[32,33]. Compelling evidence from rodents indicates that corticostriatal connectivity conveys essential information for BCI-based operant conditioning[21]. Previous evidence points to the role of the BgCN in intuitive control of behaviors[22,34], which may be called the feel strategy.

To investigate brain activity and connectivity during self-regulation, electroencephalography (EEG)-based BCI can be combined with functional magnetic resonance imaging (fMRI). Hinterberger et al.[33] conducted a pioneering concurrent BCI-MRI study with a sparse sampling method, revealing roles for both the NfCN and BgCN in self-regulation. More recently, a few concurrent BCI-fMRI studies reported the neural signature of motor imagery[32] and sense of control[35]. These studies also indicated that cortical and subcortical regions are activated during a BCI-related task, but how the NfCN[18] and BgCN[21] jointly or distinctly contribute to self-regulation remains unknown. The NfCN and BgCN are not independent but, rather, are interconnected. Recent evidence indicates that part of the striatum may serve as a hub for different BgCN networks[36], requiring revision of the canonical theory of parallel and largely segregated BgCN circuits[37]. Intriguingly, the connectivity of the striatum reflects individual variability in brain functions[38], which potentially accounts for the interindividual variability in self-regulation.

Here, we hypothesized that the intuitive BgCN and logical NfCN would play distinct roles in self-regulation during BCI. We also tested if the dorsal striatum serves as a hub interconnecting the NfCN and BgCN. We show that the relative involvement of NfCN and BgCN in self-regulation reflects interindividual differences in BCI performance.

## Results

**BCI task.** Twenty-six healthy adults participated in the concurrent BCI-fMRI experiment. Each trial began with a presentation of a horizontal bar at the bottom of the screen indicating the left target (LT), the right target (RT), or a rest (Fig. 1a). A falling cursor was then displayed for 4 s, during which its horizontal positioning was controlled by the laterality of the alpha-band (9.5–12.5 Hz) sensorimotor rhythms (SMRs) computed from electrodes C3 and C4[23]. The BCI task was to move the falling cursor horizontally to hit the target by modulating SMR laterality. For the RT and LT trials, participants were instructed to use first-person kinesthetic imagery of finger-thumb oppositions with the right and left hands, respectively. First-person kinesthetic imagery refers to a task by which the participants imagine themselves performing an action with an associated proprioceptive (not visual) sensation[39]. The rest trials served as a perceptive control, during which participants were asked to pay attention to the falling cursor without performing motor imagery. After each trial, participants were briefly notified about whether the trial was a hit or a miss (outcome period).

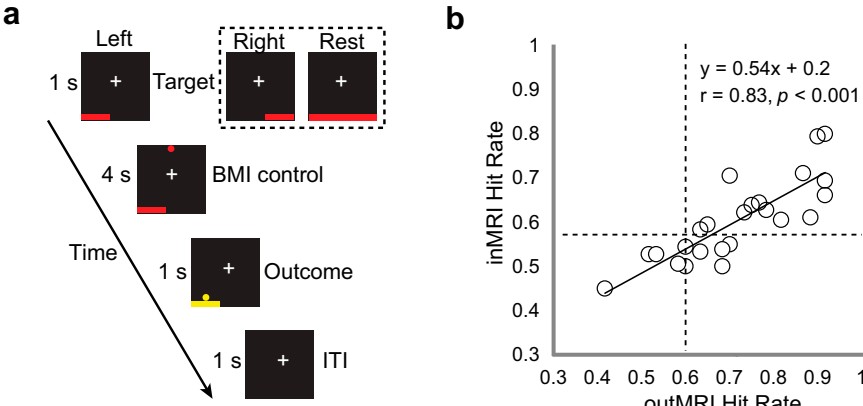

**Fig. 1 Experimental design and correlation between hit rates for sessions outside and inside MRI. a** An example of a left trial, with right and rest targets shown in the dashed box. ITI, intertrial interval. **b** Hit rates for sessions outside the MRI (outMRI) are plotted against those inside the MRI (inMRI) for each participant. On average, the inMRI hit rate (mean ± SD, 0.60 ± 0.09) was significantly lower than the outMRI hit rate (0.72 ± 0.14), as determined using a paired t test ($t_{(23)} = 6.9$, $P < 0.001$). However, hit rates were strongly correlated between the inMRI and outMRI sessions ($r = 0.83$, $P < 0.001$). Dashed lines indicate the minimum significant hit rate (out of 60 trials per task for outMRI and 90 trials per task for inMRI; $P < 0.05$ for both). Solid line indicates the linear regression.

**BCI performance**. The BCI experiment was performed over two sessions, including one practice session that was performed outside the MRI scanner while seated in a chair (outMRI) and one session that was completed inside the MRI scanner during concurrent BCI-fMRI acquisition (inMRI). After excluding data from two participants with excessive EEG artifacts, we evaluated the hit rates for the remaining 24 participants. Hit rates were calculated as the number of times the cursor hit an LT or RT divided by the total number of LT and RT trials. Twenty participants controlled the BCI significantly better than chance during the outMRI session, and 14 participants also did so during the inMRI session ($P < 0.05$, two-tailed exact binomial test; see Supplementary Table 1). The hit rates varied across participants during both the inMRI session (mean = 0.70 ± 0.13, range = 0.47–0.91) and out-MRI session (mean = 0.60 ± 0.09, range = 0.45–0.80), but these were strongly correlated ($r = 0.83$, $P < 0.001$; Fig. 1b). The offline analyses of the inMRI EEG data and extracted spectral amplitude (2–23 Hz, 3-Hz bins) showed that event-related desynchronization (ERD) of SMRs occurred mostly within the 11-Hz bin (9.5–12.5 Hz) (Supplementary Fig. 1). The SMR ERD was contralateral to the BCI target and was correlated with BCI performance (RT: $r = 0.66$, $P < 0.001$; LT: $r = -0.69$, $P < 0.001$). No statistical differences were found between the LT, RT, and rest trials for any electrooculogram or electromyographic electrode channels (repeated measures ANOVA, $P > 0.1$ for each channel), which indicates that BCI performance was not influenced by overt eye or body movements (see Supplementary Note 1 for details).

**Cortical and subcortical activity during BCI control**. We first examined whole-brain fMRI signal changes during the RT and LT trials (random-effects model analysis, $n = 24$; $P < 0.05$ cluster-level family-wise error [FWE] corrected). Compared with that during the rest trials, the self-regulation condition induced widespread activity involving both the NfCN and the BgCN, including the bilateral premotor (PM)-SMA, SPL, IPL, dlPFC, AIC, visual areas, lateral occipital cortex, basal ganglia, thalamus, and posterolateral cerebellum (lobule VI) (Fig. 2 and see Supplementary Data 1 and Table 2 for details). In particular, the primary motor cortex (M1), visual areas, and anteromedial cerebellum (lobule V) showed lateralized activity regarding the RT and LT tasks. Unexpectedly, ipsilateral M1 showed negative signal changes contributing to the SMR laterality, whereas contralateral M1 showed equivocal activity, which was not different from that of the rest trials.

**Striatal activity and corticosubcortical connectivity supporting successful BCI control**. To clarify the mechanisms underlying self-regulation for BCI control, we explored activity throughout the brain correlated with hit trials. Only the bilateral posterior putamen demonstrated greater activity during hit trials than during miss trials ($t = 3.48$, $P < 0.05$ cluster-level FWE corrected; Fig. 3a). Conversely, cortical motor areas such as the SMA tended to show less activity during hit trials than during miss trials (Fig. 3a and see Supplementary Data 2 for details). We further examined whether this hit-associated activity of the posterior putamen was present during the BCI control period or during processing of information regarding the hit/miss outcome presented at the end of each trial. The ventral striatum, which is implicated in reward and motivational processing[40,41], was chosen as a control striatal subsector. To assess the time course of brain activity, we identified volumes of interest (VOIs) in the dorsal putamen connecting with the motor cortices (motor putamen) and in the ventral striatum connecting with the orbitofrontal cortex, as defined by diffusion MRI[42]. We found greater activity in the motor putamen for the hit trials than for the

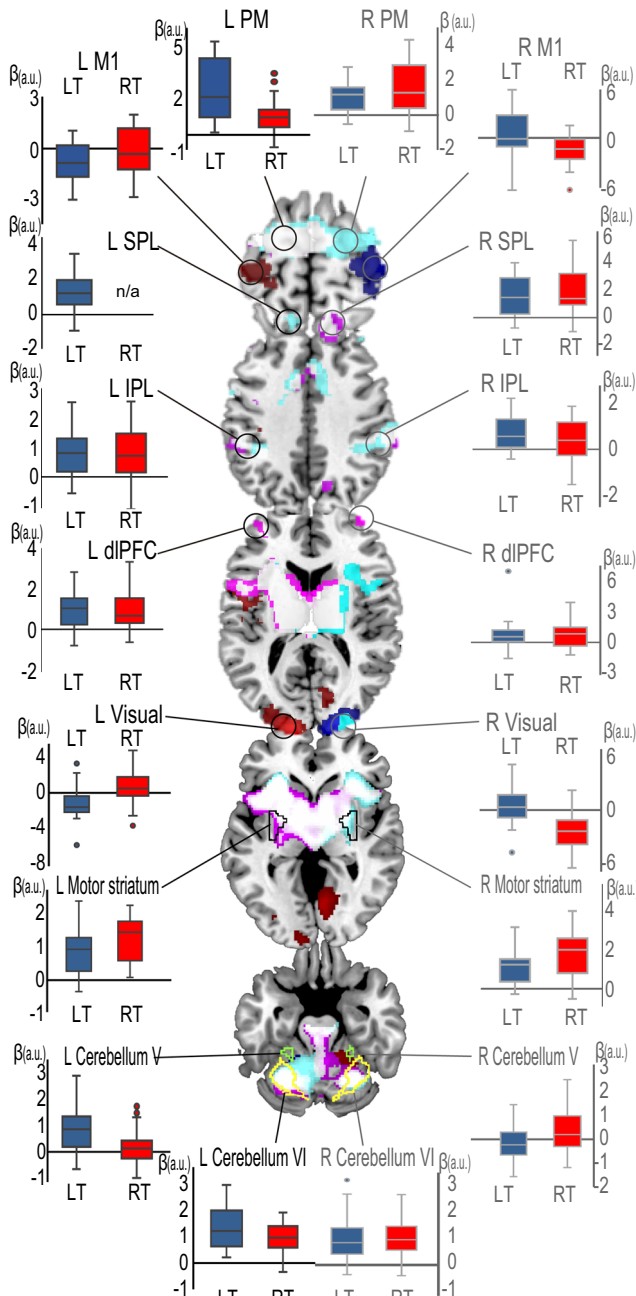

**Fig. 2 Task-related signal changes for the BCI task compared with signals at rest.** Red and blue areas denote BCI-related activity during right target (RT) and left target (LT) tasks, respectively, which differed significantly when comparing the two tasks. Magenta- and cyan-colored areas represent nonlateralized activity during RT and LT tasks, respectively, and white areas represent their overlap. PM premotor area, M1 primary motor area, SPL superior parietal lobule, IPL inferior parietal lobule, dlPFC dorsolateral prefrontal cortex, a.u. arbitrary units. The center lines of the boxplots indicate the medians, box limits indicate the lower and upper quartiles, the whiskers represent 1.5 times interquartile range, and circles indicate the outliers.

miss trials during the BCI control period ($t_{(21)} = 2.38$, $P = 0.027$, paired $t$-test), but not during the outcome presentation period ($t_{(21)} = 1.49$, $P = 0.151$) (Fig. 3b and see Supplementary Data 3 for details). By contrast, activity in the ventral striatum was comparable during the control periods ($t_{(21)} = 0.91$, $P = 0.375$) with a trend toward difference during the outcome period ($t_{(21)} = 1.86$, $P = 0.077$). This finding was supported by the analysis of the peak

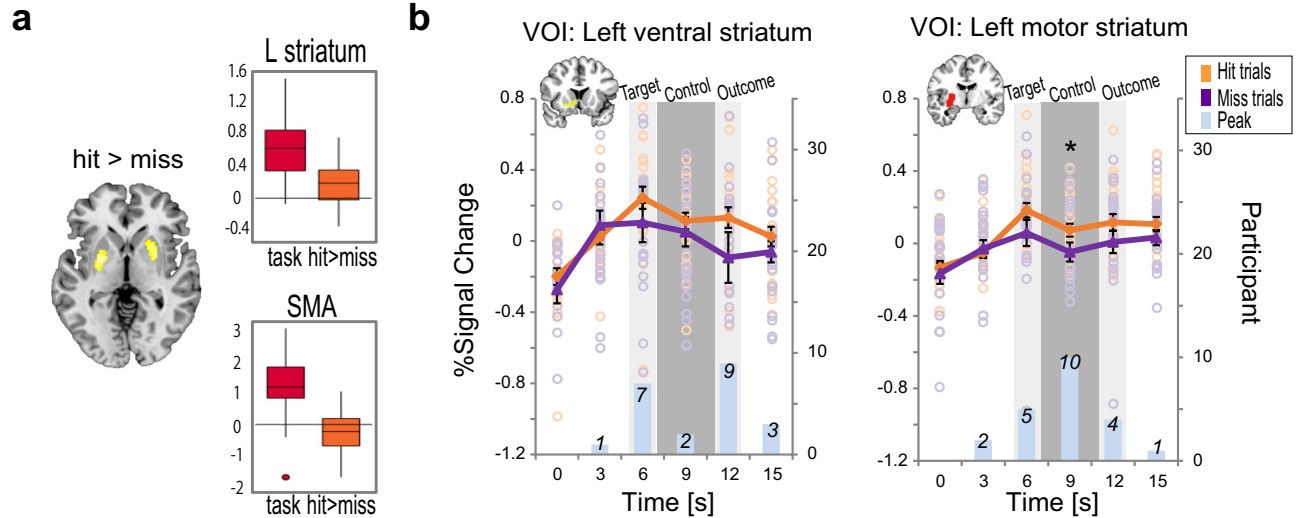

**Fig. 3 Areas of greater activation during the hit trials than during the miss trials. a** Only the bilateral dorsal striatum (yellow) showed greater BCI-related activity during the hit trials than during the miss trials (*P* < 0.05 cluster-level FWE corrected). Supplementary motor areas (SMA) showed greater BCI-related activity during the miss trials than during the hit trials. The center lines of the boxplots indicate the medians, box limits indicate the lower and upper quartiles, the whiskers represent 1.5 times interquartile range, and circle indicates the outlier. **b** Signal change time courses for the left ventral and left motor striatum for the hit (orange) and the miss (purple). The number of participants who had the peak of the hit-minus-miss activity is shown in light bule bars. Both the ventral and motor striatum, defined by diffusion MRI, showed BCI task-related activity. Only motor striatum exhibited greater activity for the hit trials than for the miss trials during the BCI control period (dark gray shaded areas, which lag 6 s behind the actual BCI control period to accommodate the hemodynamic delay). The peak of the hit-minus-miss activity is formed in the BCI control period in the motor striatum but not in the ventral striatum. Error bars indicate the standard error of the mean.

timing of the hit-minus-miss activity in the motor and ventral striatum. The peak of the hit-minus-miss activity was mainly formed during the BCI control period in the motor striatum whereas that was observed during either the target presentation period or the outcome presentation period in the ventral striatum. The timing of the peak formation was significantly different between the motor striatum and the ventral striatum (*P* = 0.046, Fisher's Exact Test). These findings suggest that motor striatal activity reflects brain mechanisms for self-regulation rather than for processing of outcome information related to a hit or miss.

These results also suggested that the motor striatum plays a role in successful self-regulation. However, it remained unclear how motor striatum activity influences BCI performance as determined by the laterality of the ERD with motor and somatosensory cortices[43]. A possibility was that the BgCN modulates SMRs. To test this hypothesis, we performed a psychophysiological interaction (PPI) analysis, using the left motor striatum as the seed (Fig. 4a). When comparing hit and miss trials, the left motor striatum showed increased connectivity (*P* < 0.05 FWE corrected) with the key nodes of the BgCN: the PM-SMA (*x, y, z* = −2, 8, 58; Z = 4.55) and globus pallidus (*x, y, z* = −24, −12, 2; Z = 5.13) extending into the thalamus. Hit-related increased connectivity with the left motor striatum was also observed in the cerebellum (*x, y, z* = −38, −52, −40; Z = 4.27) and visual cortex (*x, y, z* = −12, −88, −6; Z = 3.92). These results indicate the involvement of the BgCN[44–46] and, possibly, the cerebellar-basal ganglia circuit[47,48] in successful modulation of SMRs. Furthermore, the laterality of ERD, the critical determinant of BCI performance, correlated with BgCN connectivity during the hit trials but not during the miss trials (Fig. 4b). These findings further corroborate that the motor striatum plays a pivotal role in the self-regulation of SMRs by modulating the connectivity within the BgCN.

**Difference of functional brain networks related to individual performance.** Thus far, we found evidence for the role of the

BgCN, but not the NfCN, in the self-regulation of SMRs. Building on previous studies on the neural mechanisms underlying BCI and NFB[23–26,49–51], we exploited interindividual differences in self-regulation to test if the BgCN and NfCN jointly or distinctly contribute to successful self-regulation.

We examined whether effective connectivity with the hit-related motor striatum correlates with individual differences in BCI performance. In the PPI analysis at the individual level, the motor striatum showed various levels of effective connectivity with not only the BgCN but also the NfCN. The motor striatum regions of good performers tended to show connectivity with the BgCN, while poor performers exhibited widespread striatal connectivity with the BgCN and the NfCN (Fig. 5a).

We quantified the extent to which motor putamen connectivity with the BgCN or NfCN correlates with individual differences in the hit rate, using least absolute shrinkage and selection operator (LASSO) regression analysis. The explanatory variables were the connectivity values between the left motor striatum and the key nodes of the BgCN and NfCN: M1, PM, SMA, thalamus, cerebellum, IPL, SPL, AIC, dlPFC, and lateral occipital cortex (see Supplementary Table 2). This LASSO model predicted individual differences in BCI performance, with an $R^2$ of 0.87. The left PM and right M1 showed positive weights, indicating stronger BgCN connectivity for good performers. Conversely, the right dlPFC, IPL, PM, and cerebellum as well as the left lateral occipital cortex and AIC showed negative weights, implicating poor performance for strong striatum-NfCN connectivity (Fig. 5b and see Supplementary Data 4 for details). These results indicated that strong striatum-BgCN connectivity coupled with weak striatum-NfCN connectivity underlies the individual differences in self-regulation.

## Discussion
We showed that activity and connectivity of the BgCN reflects controllability of a BCI at both within-individual (hit vs. miss) and interindividual levels, providing evidence that BgCN

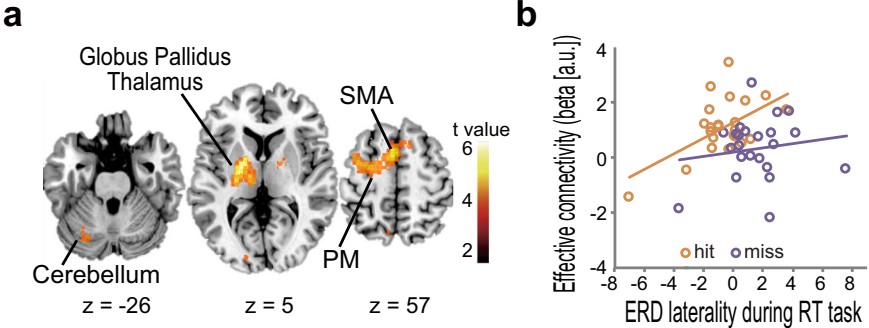

**Fig. 4 Motor cortex-basal ganglia connectivity correlates with successful BCI control. a** Hit-related effective connectivity with the left motor striatum increased in the cerebellum, globus pallidus, thalamus, and cortical motor areas, including the supplementary motor area (SMA) and dorsal premotor cortex (PM). **b** Effective connectivity between the motor striatum and SMA correlated with the laterality of event-related desynchronization (ERD) during the hit trials ($r = 0.56$, permutation test $P = 0.012$) but not during the miss trials ($r = 0.15$, $P = 0.497$) of the right-target (RT) task. Each circle represents data from one individual.

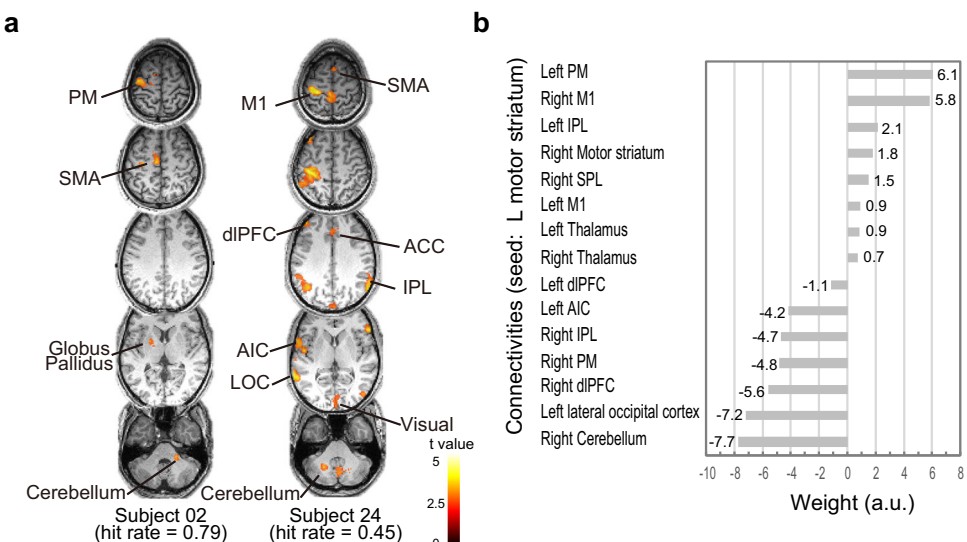

**Fig. 5 Effective connectivity with dorsal striatum represents individual differences in BCI performance. a** Effective connectivity analysis in two representative participants. Subject 02, who had good BCI performance (hit rate of 0.79), showed BgCN connectivity only, whereas subject 24, who had poor BCI performance (hit rate of 0.45), showed NfCN connectivity in addition to BgCN connectivity. **b** The LASSO regression revealed that good BCI performance correlated with BgCN connectivity, whereas it was negatively affected by connectivity with NfCN, including the lateral occipital cortex, anterior insula cortex (AIC), dorsolateral prefrontal cortex (dlPFC), and inferior parietal lobule (IPL). Weights of the left IPL, M1, dlPFC, right motor striatum, superior parietal lobule (SPL), and bilateral thalamus were less than 3.0, which represents a weak correlation.

supports the self-regulation of SMRs. Rather unexpectedly, the recruitment of the NfCN, which might reflect attempted top-down control over a BCI/NFB task[18], had a detrimental impact on self-regulation.

Consistent with previous work[32,35], the BCI task used in this study recruited motor-related cortical and subcortical areas. Contralateral M1 activity did not differ between the task and rest periods (see Fig. 2); this finding agrees with previous studies showing equivocal M1 activity during motor imagery[52,53]. However, our finding of a decrease in ipsilateral M1 activity during BCI seems novel to the best of our knowledge, and requires some explanation. During unilateral hand movement, ipsilateral M1 activity can be suppressed below a resting baseline[54]. This phenomenon is usually interpreted as the manifestation of interhemispheric inhibition. However, the concept of interhemispheric inhibition does not explain the present finding, because ipsilateral M1 activity decreased without increases in contralateral M1 activity. We propose that, in addition to the increase in neural/synaptic activity (activation), suppression of

activity below that seen at the rest baseline (i.e., deactivation) also contributes to BCI task control. This interpretation is based on the following considerations. Synchronized SMRs are a signature of a deactivated or idling motor cortex[43]. Thus, a downregulation of ipsilateral motor area activity to levels lower than those during the rest periods should correlate with ipsilateral SMR synchronization. In our study, increasingly synchronized SMRs in the ipsilateral motor cortex enhanced ERD laterality, yielding better BCI control (Supplementary Fig. 1c). Moreover, ipsilateral M1 connectivity with the striatum correlated with the individual differences in self-regulation, supporting the effectiveness of the ipsilateral deactivation strategy through the BgCN. This hypothesis should be tested in future work, since the downregulation of brain activity below a given baseline is considered an idiosyncratic strategy for BCI control[22].

A key finding from the present study is that motor striatal activity correlated with BCI performance. Striatal activity might reflect processing of the outcome stimuli, corresponding to consummatory processes during the receipt of a reward[40,41].

Although we did not use explicit incentives such as monetary rewards, striatal activity can be elicited merely by positive reinforcement[55]. However, we considered this consummatory motor striatal activity unlikely, because hit-related motor striatal activity occurred during the BCI control period but not during the outcome period. The hit-minus-miss activity formed different peaks between the ventral striatum and motor striatum. Specifically, the peak was formed during the BCI control period in the motor striatum and during the target and outcome presentation periods in the ventral striatum. The striatal activity also correlated with the trial-by-trial variation of the ERD laterality, further supporting this idea (Supplementary Fig. 2). These findings indicate that the motor striatum influences the formation of ERD laterality critical for the control of the present BCI. The ventral striatum, which underlies consummatory behavior[41], exhibited robust BCI task-related activity irrespective of the outcome. This indicates that the ventral striatum supports the motivation necessary to complete a BCI task[56] regardless of whether the response is a hit or miss.

Current BCIs often require a learning period before users can achieve adequate control[22,57,58]. In the present study, BCI control did not improve over the short experimental period during fMRI; thus the hit-related striatal activity cannot be ascribed to BCI learning. Still, we consider that the striatal activity might relate to behaviors categorized as implicit and unsupervised learning based on trial-and-error experiences[31,59]. Indeed, previous studies found striatal activity during intuitive mental processes for trials and errors[33,60,61]. Therefore, the motor striatum may serve self-regulation through an intuitive feel type of strategy.

Increased striatal activity during the hit trials was accompanied by increased effective connectivity with important nodes of the BgCN (Fig. 4a). The BgCN constitutes a semiclosed loop implicated in a variety of functions, including behavioral selection and switch, procedural learning, and motor and cognitive vigor[44,45]. The striatum may switch relevant networks according to the behavior[62,63]. We also found that BgCN connectivity correlated with ERD laterality, which was the critical determinant of BCI performance (Fig. 4b). This finding provides evidence that the striatum is involved in modulating brain rhythms through the BgCN. Therefore, the striatum needs to be included as an important module of the classic circuit for the generation of SMRs: the thalamocortical circuit[64,65]. The goal-directed modulation of SMRs may relate to the striatum's role in creating a response bias in the cerebral cortex during demanding tasks[46,66].

During the hit trials, the motor striatum also showed increased effective connectivity with the cerebellum, which also exhibited substantial BCI task-related activity. This suggests that the basal ganglia interact with the cerebellum for successful BCI control. Anatomical evidence indicates direct and reciprocal cerebellar-basal ganglia circuits via the thalamus[47,48], and these circuits may thus contribute to BCI control via the thalamic region, as revealed by the PPI analysis.

We instructed the participants to use a motor imagery strategy to modulate the laterality of SMRs, raising a question of whether the present results reflected performance of self-control or motor imagery. Previous studies suggested that activity of the premotor-parietal cortical areas (PMd/SMA, SPL and dlPFC) underlies successful motor imagery performance when motor imagery requires conscious tracking of imagined contents[52]. Consistently, Halder et al. reported that the number of activated voxels in PMd/SMA during motor observation and imagery correlated with BCI performance[24]. Zich and colleagues pointed out the role of sensorimotor areas as a signature of motor imagery[32]. Hence, the substrates of conscious motor imagery performance are likely situated in the fronto-parietal cortical network.

The motor striatum activity was specifically enhanced in the successful BCI trials whereas fronto-parietal cortical network including SMA was not correlated with the BCI hit or miss (Fig. 3a). People with Parkinson's disease in which motor striatum is impaired due to dopamine deficiency show poor BCI performance[67] whereas they do not show poor performance of conscious motor imagery unless the speed demand was imposed[42]. In the present study, good BCI performers showed the motor striatum linked functionally with the typical motor network (PMd/SMA, cerebellum, SPL) while the motor striatum of the poor BCI performers had a functional link not only with the motor networks but also with the ACC, AIC, dlPFC, lateral occipital cortex, and IPL involved in cognitive control. Therefore, good BCI performers might be proficient in BCI self-regulation since their striatum adequately selected the motor network. In contrast, poor performers showed poor selectivity of the network through the motor striatum summoning both the cognitive and motor networks. This phenomenon may relate to the disinhibition of network seen in exploratory behaviors[68]. Hence, the contribution of the motor striatum to BCI control was not just the result of successful motor imagery but rather as a kind of hub necessary for successful self-regulation.

The present results demonstrate that the BgCN and NfCN play different roles in self-regulation. Good BCI performers exhibited strong striatal connectivity with the BgCN, which included regions that subserve an implicit or bottom-up strategy of behavioral control[31,59]. BCI control may require striatal functions to modulate activity in the cortical areas to lateralize ERD. By contrast, the LASSO regression analysis revealed that NfCN connectivity was detrimental for BCI performance. Poor BCI performers exhibited stronger connectivity with the NfCN, including the dlPFC, IPL, and lateral occipital cortex. This suggests that the use of an effortful top-down or cognitive strategy results in poor BCI performance. In other words, subjects who exhibited poor BCI control might have adapted a think strategy to control SMRs declaratively or explicitly according to the task instructions about motor imagery. The improvement in BCI performance through training may involve shifts in which neural substrates are recruited, from those underlying cognitively demanding control (NfCN: think) to those related more to intuitive[34] and automatic[22] control (BgCN: feel). The employment of these two strategies at an early stage of learning may explain the interindividual differences observed in this study.

The present findings indicate that it is better to feel than to think to modulate SMRs as a BCI control signal. Researchers should thus tell BCI experiment participants "Don't think. Feel! Don't concentrate on the target but concentrate on the feeling from the fingers." Our findings indicate that altering the instructions given to participants will promote the implementation of an effective strategy and thus reduce interindividual differences in BCI controllability.

## Methods

**Participants**. Twenty-six healthy participants (12 female; mean age ± standard deviation [SD], 22.4 ± 2.9 years) participated in this study. Each participant performed two outMRI runs and three inMRI runs, which were completed on different days. All participants took part in an outMRI study before the inMRI study[23]. All participants were right-handed, as assessed using the Edinburgh Handedness Inventory[69], had normal or corrected-to-normal vision, reported no history of neurological or psychological disorders, and had no prior BCI experience. Written informed consent was obtained from all participants before participation, according to the study protocol that was approved by the institutional review board of the National Center of Neurology and Psychiatry, Tokyo, Japan. After visual inspection, data from two participants were discarded due to excessive movement-related EEG artifacts during fMRI; hence, data from 24 participants were analyzed.

**Simultaneous EEG-fMRI acquisition**. EEG was used as a BCI control modality because of its wide application in the field of neuroprosthetic control and neurorehabilitation[9–11]. A prototypical BCI was employed that uses SMRs, which arise from M1 and somatosensory cortices[32,33,35,43]. fMRI was used to measure neural/synaptic activity throughout the brain, including NfCN and the BgCN. Blood oxygen level-dependent (BOLD) fMRI acquisition was performed with a 3-T MRI scanner (Magnetom Trio; Siemens, Erlangen, Germany) using a T2*-weighted, gradient echo, echo planar imaging sequence (repetition time, 3 s; echo time, 30 ms; flip angle, 90°; voxel size, 3.0 mm³; number of slices, 42). A total of 262 scans were acquired for each run. The first 45 scans were dummy scans to minimize the transient effects of magnetic saturation and to initialize the artifact correction and BCI classifier algorithms.

Electrophysiological data were simultaneously acquired using MRI-compatible amplifiers (BrainAmp MR plus; Brain Products, Gilching, Germany) and a customized EEG cap (BrainCap MR; Brain Products)[70]. The EEG cap consisted of 13 electrodes; nine were positioned over the sensorimotor area (F3, F4, C3, C4, Cz, P3, P4, T7, and T8), one over the left eye (Fp1), one as the ground electrode (AFz), one as the reference electrode (FCz), and one attached by a 35-cm lead and placed on the back to record the electrocardiogram. Total impedances were kept at <15 kΩ. Electromyograms over the left and right thenar muscles and horizontal electrooculograms were also simultaneously acquired. Data were sampled at 5000 Hz and filtered with 0.1-Hz high-pass and 250-Hz low-pass hardware filters.

**Online EEG artifact correction**. To provide online BCI classification and online feedback, MRI artifacts incurred on the EEG data were corrected online. According to methods reported by Allen and colleagues[71], artifact correction algorithms were written in MATLAB R2007b (MathWorks, Natick, MA, USA) that operated in conjunction with the BCI software. The system first corrected gradient artifacts, which are millivolt-scale distortions in EEG that are caused by the switching gradient magnetic fields of MRI. Exploiting the fact that gradient artifacts are mostly stationary and phase locked to the repetition timing of a sequence, mean gradient artifact templates were calculated for each channel and subtracted from the incoming EEG.

After gradient artifact correction, the system then corrected for ballistocardiogram artifacts, which are microvolt-scale deflections in the EEG resulting from micromovements of the head that are induced by the pulsatile acceleration of blood through the aortic arch[72] and possibly also by the expansion and contraction of scalp arteries[71]. The R peak of electrocardiogram precedes artifacts by approximately 200 ms; this information was used to create a mean ballistocardiogram template over the R-R interval for each channel, which was subtracted from the EEG, similarly to the gradient artifact correction process.

To limit the effects of gross and ballistocardiogram-related head movements, a custom-made vacuum cushion[73] was placed around the participant's head that conformed to the space inside the MRI head coil. Finally, to reduce remnant scanner-related noise and baseline drift, a 12th-order elliptic bandpass filter (1–23-Hz bandpass, 0.1-dB passband ripple, and 20-dB attenuation) was applied after ballistocardiogram artifact correction. Signals were then downsampled to 500 Hz for further processing.

**Brain-machine interface control and feedback**. For the BCI system used in this study, visual stimuli, feature extraction, and classification were all performed using the BCI2000 software platform[74]. Participants were asked to perform two motor imagery tasks: imagery of finger-thumb opposition with the left and right hands, and a baseline rest task. For the imagery tasks, participants were instructed to use, to the best of their ability, a first-person perspective and kinesthetic rather than visual imagery[75]. Participants also overtly practiced the movements before the start of the experiment.

Tasks were cued using visual stimuli (Fig. 1a) that were projected onto a mirror attached to the MRI head coil. For each trial, a rectangular target appeared in the lower left, lower right, or entire bottom portion of the display, which cued the LT, RT, or rest task, respectively. After 1 s, a cursor appeared at the top center of the screen and immediately began falling at a constant rate, such that it would reach the bottom in 4 s. During imagery trials, participants were tasked with using motor imagery to control the horizontal positioning of the cursor so that it would hit the target at the bottom. During rest trials, participants were asked to passively watch the display and refrain from performing the imagery tasks. When the cursor reached the bottom, a 1-s interval ensued, during which the cursor and target either turned yellow in the case of a hit trial or remained unchanged. The next trial began after a 1-s intertrial interval with a blank screen.

Trials were organized into blocks, with each block containing three trials of the same task. A run consisted of 11 pseudorandom permutations of LT and RT blocks interleaved with 12 rest blocks. Each run began and ended with a rest block. The first block of each task was used for classifier calibration and was discarded, leaving a total of 30 LT, 30 RT, and 33 rest trials per run. To evaluate BCI performance, the hit rate was calculated as the number of times the cursor hit the left or right target divided by the number of imagery trials in each run for each participant. The hit rate was calculated over all three runs, and the overall significance was compared with chance (58%, $P < 0.05$, two-tailed exact binomial test). BCI performance was pooled from three inMRI runs, as no differences in hit rate were found between runs ($F_{(1.9, 43.5)} = 1.86$, $P = 0.17$).

**Feature extraction and cursor control**. After undergoing noise reduction and downsampling, feature extraction and classification were performed to provide BCI control signals[74]. Electrodes over the sensorimotor area (F3, F4, C3, C4, Cz, P3, P4, T7, and T8) were re-referenced to large Laplacian derivations for C3 and C4. Spectral amplitudes for C3 and C4 were then computed using autoregressive estimation[76,77], with a window length of 500 ms and bin width of 3 Hz. For all participants, spectral amplitudes were selected from the 9.5–12.5-Hz bin for feature extraction, which allowed the BCI to be controlled with SMR desynchronizations related to motor imagery[43]. The fixed 9.5–12.5-Hz bin was selected according to the results of the previous outMRI study showing that it is effective for controlling the BCI[23]. The use of a single frequency band also made it easier to analyze and interpret all participants' data as a group. Harmonic noise detected from the MRI scanner precluded the inclusion of beta activity (13–30 Hz), which can be used for BCI control.

A control signal for cursor movement was computed from the interhemispheric difference (C4 minus C3) during ERD. At each time point, the control signal was normalized to the zero mean and unit variance based on data from the previous two trials of all three tasks. This normalization provided a linear classifier for the RT and LT tasks. When the C4 spectral amplitude decreased relative to the C3 spectral amplitude (i.e., was desynchronized), the cursor accelerated to the left. Conversely, sufficient C3 desynchronization resulted in rightward cursor movement.

**fMRI data preprocessing**. All fMRI data preprocessing and analyses were performed using SPM8 (Wellcome Trust Center for Neuroimaging, London, UK). The functional images underwent slice-timing correction and spatial realignment. The realigned images were then normalized to the Montreal Neurological Institute stereotactic space using the standard echo-planar imaging template in SPM8. Finally, the normalized images were spatially smoothed using a Gaussian kernel of 6-mm full-width at half-maximum.

**Statistics and Reproducibility**. For the first-level analysis, within-subject task effects were examined by including LT, RT, and rest as conditions plus head motion parameters in a general linear model. Onset time and duration of fMRI data corresponded to the 4-s BCI control intervals, during which the cursor was moving and participants were controlling its position. Performance-related effects were examined by including two binary parametric modulators corresponding to the hit-and-miss trials for the LT and RT tasks.

The second-level analysis revealed greater bilateral dorsal striatum activity during the hit trials than during the miss trials (Fig. 3a). We calculated the difference in activity time-course between the hit and miss trials in the ventral putamen and the motor putamen in each participant. We then identified the peak of the hit-minus-miss activity in the 5 points (3 s, 6 s, 9 s, 12 s, and 15 s). Considering the ~6 s delay of the hemodynamic responses, the 6-s time-point corresponded to the target stimulus presentation, 9-s time-point to the BCI control, and the 12-s time-point to the outcome presentation. The difference in the distribution of the peak was compared across the ventral putamen and the motor putamen, using a Fisher's Exact Test for the 2 VOIs (motor striatum and ventral striatum) and the 5-time points contingency tables (Fig. 3b). PPI analysis was performed to examine functional coupling of the dorsal striatum with other regions throughout the brain (Fig. 4). The dorsal striatum served as a seed region, with the BOLD time series applied as a physiological variable, whereas the parametric modulators from the hit/miss model were used as psychological variables. These psychophysiological variables and their interaction were then applied in a model to identify areas that were functionally coupled with the seed region.

For all designs, data were high-pass filtered (1/128-Hz cutoff) to remove low-frequency drift, and realignment parameters acquired during preprocessing were included to regress out head movement artifacts. Second-level, random-effect model group analyses were then performed using contrast-weighted beta images from the first-level analysis. The height threshold at the voxel level was set to a $P$ value of <0.001, and FWE correction at the cluster level ($P < 0.05$) was performed using SPM8's implementation of random field theory.

**Striatum parcellation and fMRI**. A map of striatal subdivisions was created using a diffusion-based subcortical gray matter classification technique[42,78,79]. In brief, diffusion tensor MR images ($b = 1000$ s/mm²) were collected from a separate group (15 volunteers, 5 female, aged $26.7 ± 10.1$ years), after obtaining written consent and approval by the institutional ethics committee[42]. Probabilistic diffusion tractography running between the whole striatum seed and the frontal cortical areas was analyzed using FSL4.1. The cortical subdivision that had the highest connectivity was identified after scaling connectivity in each cortical region relative to the total for each voxel in the entire striatum. The ventral striatal VOI connecting with the orbitofrontal cortex and the motor striatal VOI connecting with M1 and Brodmann area 6, including the SMA and PM, were used. BOLD time series data were extracted from these two striatal VOIs. To assess the time series, data from the BCI control periods just after the rest trials were classified into hit and miss trials. In this analysis, data from two participants were excluded ($n = 22$): one participant had an extreme performance (no miss trials for the selected condition) and the

other participant had bumpy BOLD time-series data (>2 SDs from the group data), possibly due to head motion.

**LASSO regression analysis of BCI performance**. To predict task performance during RT, LASSO regression (sklearn.linear_model; http://jupyter.org/) was applied using connectivity with the left motor striatum as an explanatory variable. The LASSO is a linear regression with an L1 norm penalty term (which thus introduces sparseness)[80]. A hyperparameter for the penalty weight was determined by 2-fold cross validation ($\alpha = 0.11$). The explanatory variables were values of connectivity with the left motor striatum, which were calculated from a 10-mm spheric VOI set at the peak coordinate of each cluster in the fMRI analysis. VOIs were set in the following 22 regions: the cerebellum, PM, SMA, AIC, M1, SPL, IPL, dlPFC, lateral occipital cortex, motor striatum, and thalamus (all bilateral). To assess model fitting, the correlation of the determination parameter was calculated ($R^2 = 0.87$).

**Reporting summary**. Further information on research design is available in the Nature Research Reporting Summary linked to this article.

## Data availability

Source data underlying figures are provided in Supplementary Data 1–4. Any additional data that support the findings of this study are available from the corresponding author upon reasonable request.

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

## Acknowledgements

Japan Agency for Medical Research and Development grants 19dm0207070s0001 and 19dm0307003h0002 (TH). Japan Society for the Promotion of Science grants KAKANHI 19H05726 and 19H03536 (TH) and KAKANHI 20H04236 (KK). Japan Science and Technology Agency grant FORESTO JPMJFR206G (KK). Charles S. DaSalla is currently affiliated with the Office of Research and Innovation, Tokyo Institute of Technology.

## Author contributions

Conceptualization: T.H. Methodology: K.K., C.S.D., M.H., and T.H. Investigation: K.K. and C.S.D. Writing—original draft: K.K., C.S.D., and T.H. Writing—review & editing: K.K., C.S.D., M.H., and T.H. Funding acquisition: T.H. and K.K. Resources: M.H. and T.H. Supervision: M.H. and T.H.

## Competing interests

The authors declare no competing interests.
