## [Peer Review File · Communications Biology]

Reviewers' comments:

Reviewer #1 (Remarks to the Author):

Kasahara and colleagues present an elegant and highly interesting study. They apply simultaneously/sequentially acquired fMRI-EEG (3T) data in 26 young (mean age 22.4 years) healthy volunteers probing the neuronal correlates for brain-machine interface control and feedback. The results indicate that the basal ganglia cortical network and the neurofeedback control network play different roles in self-regulation.

The manuscript is highly recommended for publication.

Reviewer #2 (Remarks to the Author):

In the manuscript, the authors performed simultaneous electroencephalography (to assess BCI control) and functional magnetic resonance imaging in healthy participants. They reported that self-regulation of cortical oscillations induced activity in the basal ganglia-cortical network (BgCN) and the neurofeedback control network (NfCN). They also showed that the relative involvement of NfCN and BgCN in self-regulation reflects interindividual differences in BCI performance.

This is a very interesting study with novel findings and well presented. I only have some minor comments:

1. The potential functional separation of the motor-associated putamen and ventral striatum in BCI control is very novel and interesting. However, the results showed in Fig. 3B also indicate that the activity pattern for these two regions are quite similar. For statistics, motor-associated putamen showed significant difference between hit and miss; whereas the ventral striatum did not. A more rigorous statistics would be a two-way ANOVA, to see if there is significant interaction between BCI performance (hit/miss) and the region (motor-associated putamen/ventral striatum).

2. Is there any across-trial correlation within participant or cross-participant correlation between the activity in the motor-associated putamen and BCI performance, or ERD laterality, or ERD in the targeted frequency band (~11 Hz) or the other motor rhythm (the beta frequency band between 20 and 30 Hz) in C3 or C4? This would also be a more direct proof about the statement on the role of motor-associated putamen on sensorimotor rhythm modulation.

3. The correlation between the ERD laterality and the connectivity between the motor striatum and SMA: the scatter plot in Fig.4D indicates that the correlation was not particularly strong. I would suggest to use permutation (permute the order of one variable and calculate the correlation, repeat this for 1000 times, to derive the baseline correlation coefficient distribution), and see the correlation is still within 5% confidence level.

Reviewers' comments:

Reviewer #1: Kasahara and colleagues present an elegant and highly interesting study. They apply simultaneously/sequentially acquired fMRI-EEG (3T) data in 26 young (mean age 22.4 years) healthy volunteers probing the neuronal correlates for brain-machine interface control and feedback. The results indicate that the basal ganglia cortical network and the neurofeedback control network play different roles in self-regulation. The manuscript is highly recommended for publication.

Response

We highly appreciate the positive comments.

Reviewer #2: In the manuscript, the authors performed simultaneous electroencephalography (to assess BCI control) and functional magnetic resonance imaging in healthy participants. They reported that self-regulation of cortical oscillations induced activity in the basal ganglia-cortical network (BgCN) and the neurofeedback control network (NfCN). They also showed that the relative involvement of NfCN and BgCN in self-regulation reflects interindividual differences in BCI performance. This is a very interesting study with novel findings and well presented. I only have some minor comments:

Response

Thank you for the positive comments overall and also for the important comments to help improve the manuscript.

Comment 1: The potential functional separation of the motor-associated putamen and ventral striatum in BCI control is very novel and interesting. However, the results showed in Fig. 3B also indicate that the activity pattern for these two regions are quite similar. For statistics, motor-associated putamen showed significant difference between hit and miss; whereas the ventral striatum did not. A more rigorous statistics would be a two-way ANOVA, to see if there is significant interaction between BCI performance (hit/miss) and the region (motor-associated putamen/ventral striatum).

Response

We agree with this comment. We did not perform the recommended analysis in the previous version because we do not usually compare the size of activity across different brain regions, which inherently have different contrast-to-noise ratios. Indeed, when we performed a two-way ANOVA (region x performance at time 6 or 9) and a three-way repeated measures ANOVA (time x region x performance) according to the reviewer's suggestion, but we failed to find interactions between the region and the performance. This was not surprising when looking at similar signal time

courses across the conditions during the BCI task. However, because we thought that the issue raised by the reviewer was important, we explored a few alternative ways to address this issue and eventually came up with the following idea. That was, we calculated the difference in activity time-course between the hit and miss trials in the ventral putamen and the motor putamen in each participant. We then identified the peak of the hit-minus-miss activity in the 5 time bins (3s, 6s, 9s, 12s, 15s; information added to Figure 3B). We then statistically compared the distribution of the hit-minus-miss peak between the motor putamen and the ventral striatum, using Fisher's Exact Test. The test demonstrated that the hit-minus-miss activity peaked at different timings across the two striatal regions. Notably, most participants had the hit-minus-miss activity peak during the BCI control period in the motor striatum. By contrast, in ventral striatum, most participants had the hit-minus-miss activity peak at the target cue presentation or the outcome presentation. This finding supports the idea that the motor striatum controls BCI, and that the ventral striatum analyzes information attached to task-relevant sensory information. The methods, results, and Figure3 have been added to main text as follows:

Method

We calculated the difference in activity time-course between the hit and miss trials in the ventral putamen and the motor putamen in each participant. We then identified the peak of the hit-minus-miss activity in the 5 points (3s, 6s, 9s, 12s, and 15s). Considering the ~6 s delay of the hemodynamic responses, the 6-s time-point corresponded to the target stimulus presentation, 9-s time-point to the BCI control, and the 12-s time-point to the outcome presentation. The difference in the distribution of the peak was compared across the ventral putamen and the motor putamen, using a Fisher's Exact Test for the 2 VOIs (motor striatum and ventral striatum) and the 5 time points contingency tables (**Fig. 3B**).

(lines 503-510)

Result

We found greater activity in the motor putamen for the hit trials than for the miss trials during the BCI control period ($t_{(21)} = 2.38$, $P = 0.027$, paired t test), but not during the outcome presentation period ($t_{(21)} = 1.49$, $P = 0.151$) (**Fig. 3B**). By contrast, activity in the ventral striatum was comparable during the control periods ($t_{(21)} = 0.91$, $P = 0.375$) with a trend toward difference during the outcome period ($t_{(21)} = 1.86$, $P = 0.077$). This finding was supported by the analysis of the peak timing of the hit-minus-miss activity in the motor and ventral striatum. The peak of the hit-minus-miss activity was mainly formed during the BCI control period in the motor striatum whereas that was observed during either the target presentation period or the outcome presentation period in the ventral striatum. The timing of the peak formation was

significantly different between the motor striatum and the ventral striatum ($P = 0.046$, Fisher's Exact Test).

(lines 141-150).

Discussion

The hit-minus-miss activity formed different peaks between the ventral striatum and motor striatum. Specifically, the peak was formed during the BCI control period in the motor striatum and during the target and outcome presentation periods in the ventral striatum. The striatal activity also correlated with the trial-by-trial variation of the ERD laterality, further supporting this idea (**Supplementary Figure 2**). These findings indicate that the motor striatum influences the formation of ERD laterality critical for the control of the present BCI.

(lines 221-227)

Figure 3 and legends

Fig. 3. Areas of greater activation during the hit trials than during the miss trials. (A) Only the bilateral dorsal striatum (yellow) showed greater BCI-related activity during the hit trials than during the miss trials ($p < 0.05$ cluster-level FWE corrected). Supplementary motor areas (SMA) showed greater BCI-related activity during the miss trials than during the hit trials. (B) Signal change time courses for the left ventral and left motor striatum for the hit (orange) and the miss (purple). The number of participants who had the peak of the hit-minus-miss activity is shown in light blue bars. Both the ventral and motor striatum, defined by diffusion MRI, showed BCI task-related activity. Only motor striatum exhibited greater activity for the hit trials than for the miss trials during the BCI control period (dark gray shaded areas, which lag 6 s behind the actual BCI control period to accommodate the hemodynamic delay). The peak of the hit-minus-miss activity is formed in the BCI control period in the motor striatum but not in the ventral striatum. Error bars indicate the standard error of the mean.

(lines 357-367)

Comment 2: Is there any across-trial correlation within participant or cross-participant correlation between the activity in the motor-associated putamen and BCI performance, or ERD laterality, or ERD in the targeted frequency band (~11 Hz) or the other motor rhythm (the beta frequency band between 20 and 30 Hz) in C3 or C4? This would also be a more direct proof about the statement on the role of motor-associated putamen on sensorimotor rhythm modulation.

Response

Thank you for the comment. We understand that presentations for the correlation between activity in putamen and performance would be a more direct proof of our idea. We actually had presented one of the suggested analyses below. We added another analysis and found consonant results.

Trial-by-trial BCI performance correlation: Actually, we had already presented the putaminal activity greater during the hit trials than the miss trials in Figure 3 of the previous version (this corresponds to activity correlated with the performance across the trials within the participants). Because the BCI performance was binary (hit vs. miss), the trial-by-trial variation of the BCI performance corresponded to the hit vs. miss. We created the hit regressor by which 1 and -1 was assigned to the hit and miss trials, respectively, and fed them into the first-level fMRI analysis in each individual. The second-level analysis from this hit regressor yielded the original Fig. 3A, which corresponded to one of the suggested analyses.

Trial-by-trial ERD laterality correlation: We used the laterality of ERD at the 11-Hz bin as a control signal of the BCI. Although the ERD laterality was directly related to the binary variable of the hit or the miss, the ERD laterality was a continuous variable (**Supplemental Figure 2A**), providing slightly different information from the hit vs. miss. Hence, we decided to analyze correlation between the trial-by-trial variation of ERD laterality and that of brain activity during the right BCI trials. The trial-by-trial variation of ERD laterality was indeed correlated with activity in the putamen, overlapping with the putaminal activity correlated with BCI performance (hit vs. miss) (**Supplemental Figure 2B**). The overlap between the BCI performance and ERD laterality correlation makes sense and enhances the statement on the role of the putamen in the sensorimotor rhythm modulation. These results have been added to Supplemental Information to support our argument as a **Supplemental Figure 2** and Discussion as follows:

Supplemental Figure 2 Trial-by-trial ERD laterality. **A.** Trial-by-trial variation of the ERD laterality in all trials for all participants. Each circle indicates the ERD laterality in each trial in each participant. The crosses indicate the mean ERD laterality in each participant. The ERD laterality was computed from raw EEG data before any weighting or calibration by BCI2000. **B.** Correlation between the trial-by-trial ERD laterality and brain activity during the BCI control period for the right target (green). This activity overlapped with the putamen that were more activated in the hit than the miss trials shown in red. The yellow areas indicate the overlap of the two activities.

Discussion

The striatal activity also correlated with the trial-by-trial variation of the ERD laterality, further supporting this idea (**Supplementary Figure 2**). These findings indicate that the motor striatum influences the formation of ERD laterality critical for the control of the present BCI. (lines 224-227)

Comment 3: The correlation between the ERD laterality and the connectivity between the motor

striatum and SMA: the scatter plot in Fig.4D indicates that the correlation was not particularly strong. I would suggest to use permutation (permute the order of one variable and calculate the correlation, repeat this for 1000 times, to derive the baseline correlation coefficient distribution), and see the correlation is still within 5% confidence level.

Response

Thank you for your very helpful comment. As you suggested, we calculated the correlation coefficient using permutation (1000 times), and the p value was still within the 5% confidence level ($p = 0.012$). This statistical result is so informative that it has been added to the line 374.

REVIEWERS' COMMENTS:

Reviewer #2 (Remarks to the Author):

The authors have addressed all my comments. I think it is an interesting paper with data and results that would be of interest to the community.